# Representation Learning on Spatial Networks

**Zheng Zhang**
Department of Computer Science
Emory University
Atlanta, GA 30322, USA
`zheng.zhang@emory.edu`

**Liang Zhao**
Department of Computer Science
Emory University
Atlanta, GA 30322, USA
`liang.zhao@emory.edu`

## Abstract

Spatial networks are networks for which the nodes and edges are constrained by geometry and embedded in real space, which has crucial effects on their topological properties. Although tremendous success has been achieved in spatial and network representation separately in recent years, there exist very little works on the representation of spatial networks. Extracting powerful representations from spatial networks requires the development of appropriate tools to uncover the pairing of both spatial and network information in the appearance of node permutation invariant, and rotation and translation invariant. Hence it can not be modeled merely with either spatial or network models individually. To address these challenges, this paper proposes a generic framework for spatial network representation learning. Specifically, a provably information-lossless and rotation-translation invariant representation of spatial information on networks is presented. Then a higher-order spatial network convolution operation that adapts to our proposed representation is introduced. To ensure efficiency, we also propose a new approach that relied on sampling random spanning trees to reduce the time and space complexity from $O(N^3)$ to $O(N)$. We demonstrate the strength of our proposed framework through extensive experiments on both synthetic and real-world datasets. The code for the proposed model is available at `https://github.com/rollingstonezz/SGMP_code`.

## 1 Introduction

Spatial data and network data are both popular types of data in modern big data era. The study of spatial data focuses on the properties of *continuous* spatial entities under specific geometry, while analysis of network data investigates the properties of *discrete* objects and their pairwise relationship.

Spanning these two data types, spatial networks is a crucial type of data structure that nodes occupy positions in a Euclidean space, where spatial patterns and constraints may have a strong effect on their connectivity patterns [6]. Understanding the mechanism of organizing spatial networks has significant importance for a broad range of fields [29], ranging from micro-scale (e.g., molecule structure [94]), to middle-scale (e.g., biological neural network [31]), to macro-scale (e.g., mobility networks [18]). Effectively learning the representations of spatial networks is extremely challenging due to the close interactions between network and spatial topology, the incompatibility between the treatments for discrete and continuous data, and particular properties

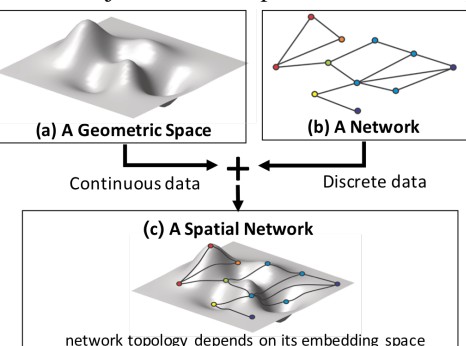

Figure 1: Spatial network contains not only the information of network topology and spatial topology but also their interaction.

35th Conference on Neural Information Processing Systems (NeurIPS 2021).

such as permutation invariant and rotation-translation invariant. Spatial networks have long been researched in the domains such as physics and mathematics, which usually extend complex networks and graph theory into spatial networks [80, 7]. They typically rely on network generation principles predefined by human heuristics and prior knowledge. Such methods usually characterize well on the aspects of the data that have been covered by the predefined principles, but not on those have not been covered[6]. However, the underlying network process in complex networks is largely unknown and extremely difficult to be predefined in simple rules, especially in crucial and open domains such as brain network modeling [81], network catastrophic failure [72], and protein folding [27].

Remarkable progress has been made towards generalizing deep representation learning approaches in spatial data and network data [96, 13, 46, 43, 28], respectively, in recent years. For spatial data, deep learning achieved significant progress in different commonly used formats such as images [53, 75, 61, 21], point clouds [34, 71, 57], meshes [85, 90], and volumetric grids [95, 66]. On the other hand, deep learning has also boosted the research of encoding graph structure on network data [46, 51, 45], and downstream applications such as recommender systems [99, 60], drug discovery [39, 22, 41, 42], FinTech [91], customer care [93], and natural language processing [62, 8, 92].

Despite the respective progress in representation learning on spatial data and network data in parallel, the representation learning for spatial networks have been largely underexplored and has just started to attract fast-increasing attention. Merely combining spatial and graph representations separately cannot handle that for spatial networks where spatial and network process are deeply coupled together [6, 68, 33]. For example, Fig. 2 shows a simple example with a pair of spatial networks, in which there are different formation rules on the edges relied on the spatial distance, that is non-distinguishable for spatial and network embedding methods, respectively.

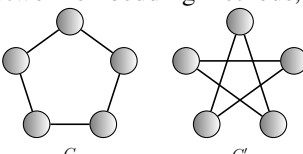

Few recent attempts have been proposed to handle representation learning on spatial networks but still suffer from key challenges: Spatial network representation learning is a problem extremely difficult to address due to several unique challenges: 1) Difficult in distinguishing the patterns that require joint spatial and graph consideration. Examples like Figure. 2 that share the same spatial and network topology, respectively, but with significantly different interaction mechanisms, are non-distinguishable to either spatial or graph methods. 2) Difficult in jointly maintaining that the learned representation is invariant to node permutation, and rotation and translation transformations. Notice that spatial and graph information confine each other which neutralizes conventional methods to have either of them. For example, although point clouds representation learning can easily preserve rotation- and translation-invariant by using spatial nearest neighbors, here in spatial networks the neighbor is confined also by graph neighbors. Such additional confinement largely harden our task. 3) High efficiency and scalability in the graph size. The confinement between spatial and graph information inevitably leads to taking into account more entities simultaneously to maintain sufficient information. The requirement to handle incremental information increases the demand for model efficiency and scalability.

Figure 2: The left figure reflects closer nodes tend to connect with each other (known as the first law of geography [79]), while the right figure reflects a spatial tele-connecting pattern where faraway nodes tend to connect. Discriminating these two spatial networks requires new method that can jointly consider spatial and network properties.

In order to address all the aforementioned challenges, this paper proposes a new spatial graph message passing neural network (SGMP) for learning the representations of generic spatial networks, with theoretical guarantees on discriminative power and various spatial and network properties, and an accelerating algorithm which adjusts to our theoretical framework. Specifically, to capture and model the intrinsic coupled spatial and graph properties, we propose a novel message passing neural network to organically aggregate the spatial and graph information. To ensure the invariance of learned representation under rotation and translation transformations, a novel way to represent the node spatial information by characterizing geometric invariant features with lossless information is proposed. To alleviate the efficiency issue, we propose a new accelerating algorithm for learning on graph-structured data. The proposed accelerating algorithm effectively reduces the time and memory complexity from $O(N^3)$ to $O(N)$, and maintains the theoretical guarantees for spatial networks. Finally, we demonstrate the strength of our theoretical findings through extensive experiments on both synthetic datasets and real-world datasets.

## 2 Related Work

**Spatial Networks**   There has been a long time of research efforts on the subjects of spatial networks [6]. In the area of quantitative geography, Haggett and Chorley discussed the relevance of space in the formation and evolution of networks, and developed models to characterize spatial networks at least forty years ago [44, 17]. New insights leading to modern quantitative solutions are gained due to the advance in complex networks [32, 89, 1, 4, 3, 20], and appears in more practical fields such as transportation networks [2, 54, 55], mobility networks [18, 24], biological networks [31, 76], and computational chemistry [38, 73, 52].

**Geometric Deep Learning.** This is a more recent domain which handles non-Euclidean structured data such as graphs and manifolds [13].
• *Geometric Deep Learning on Manifolds.* There is a large body of research efforts of generalizing deep learning models to 3D shapes as manifolds in the computer graphics community. Many works have been conducted to find a better approach to generalize convolution-like operations to the non-Euclidean domain [65, 10, 74, 98, 64, 59]. J. Masci *et at.* proposed the framework of generalizing convolution neural network paradigm to manifolds by applying filters to extract local patches in polar coordinates [65]. Litany *et at.* [59] proposed FMNet to learn the dense correspondence between deformable 3D shapes.
• *Geometric Deep Learning on Graphs.* The earliest attempts we are aware of to generalize neural networks to graphs are attributed to M. Gori *et at.* [40]. More recently, a number of approaches encouraged by the success of convolutional neural networks [53] have attempted to generalize the notion of convolution to graphs. One important stream of convolution graph neural networks is spectral-based, where emerges after the pioneering work of Bruna *et at.* [11] which based on the spectral graph theory. There have been many following works [48, 25, 51, 56]. Another stream of work define graph convolutions as extracting locally connected regions from the graph [30, 58, 69, 45, 97, 67]. Many of these works were formulated in the family of message passing neural networks [38] which apply parametric functions to a node and its proximities, and use pooling operations to generate features for the node. Efficiency and scalability for deep graph learning is very important especially for large graphs and higher-order operations, which triggers research on accelerating GNNs [45, 16, 15]. Hamilton *et at.* [45] first introduced sampling scheme on neighborhood nodes to restrict the size. Chen *et at.* [16] proposed a method which samples vertices rather than neighbors. However, none of these works can guarantee the sampled graph is connected.

**Deep Learning on Spatial Data.** Deep learning has also boosted the study on spatial data. Significant progress has been achieved on deep learning on images since AlexNet [47, 75]. For 3D point clouds, PointNet [71] is a pioneering work which addressed the permutation invariance by a symmetric function. PointCNN [57] transforms the input points into a latent and potentially canonical order by a $\chi$-conv transformations. Volumetric-based methods usually apply a 3D Convolution Neural Network (CNN) to 3D grids [95, 66]. Wang *et at.* [85] first performed shape segmentation on 3D meshes by taking three low-level geometric features as its input.

Despite the success of generalizing deep learning to network and spatial data separately, there has been relatively little work that simultaneously characterize both of them and their interaction. Previous models such as [38, 73, 52] are domain-specific, [73, 52] treat spatial networks as point clouds which ignores the influence of network structure, and [87, 86] consider POI (Point of Interest) categories, hence such concept graphs are not physically embedded in a geometric space. In addition, existing works [88, 23] typically utilize the off-the-shelf deep neural networks with Cartesian coordinates as inputs and a large amount of rotation-and translation-augmented data, which is computationally expensive and lacks theoretical guarantee of rotation- and translation-invariant on the representation. To the best of our knowledge, our proposed method is the first generic framework of spatial network representation learning that handles substantial properties of rotation- and translation-invariant and the interplay between spatial and graph patterns with a theoretical guarantee.

## 3 Preliminaries

In this section, we first formalize spatial networks, and the problem of representation learning on spatial networks, then we introduce the challenges in order to solve this problem.

Spatial graphs (also known as spatial networks [6]) are networks for which the nodes and edges are embedded in a geometric space. Spatial networks is ubiquitous in real world, such as molecular graphs [94], biological neural networks [31], and mobility networks [18], where the spatial and

network properties are usually coupled together tightly. For example, chemical bonds are derived from spatially close atoms, and fiber nerves tend to connect neurons close to each other. A spatial network is typically defined as $S = (G, P)$, where a graph $G = (V, E)$ denotes the graph topology such that $V$ is the set of $N$ nodes and $E \subseteq V \times V$ is the set of $M$ edges. $e_{ij} \in E$ is an edge connecting nodes $v_i$ and $v_j \in V$. $P$ denotes the spatial information that is expressed as a set of points $P = \{(x_i, y_i, z_i) | x_i, y_i, z_i \in \mathbb{R}\}$ in Cartesian coordinate system, such that for a node $v_i \in V$, its coordinate is denoted as $(x_i, y_i, z_i) \in P$. Permutation invariance are crucial to graph structured data [96]. The collections of *permutation-invariant functions* on graph-structured data is defined so that $f(\pi^\dagger S \pi) = f(S)$, for all $\pi \in S_n$, where $S_n$ is the permutation group of $n$ elements. Rotation and translation invariance are in natural and common requirements for spatial data [36, 52]. The collections of *rotation- and translation-invariant functions* on spatial networks is defined so that $f(G, \mathcal{T}(P)) = f(G, P)$, for all $\mathcal{T} \in \text{SE}(3)$, where $\text{SE}(3)$ is the continuous Lie group of rotation and translation transformations in $\mathbb{R}^3$.

The main goal of this paper is to learn the representation $f(S)$ of spatial network $S = (G, P)$, with the simultaneous satisfaction of strong discriminative power and the aforementioned significant symmetry properties, which is a problem extremely difficult to address due to several unique challenges: 1) Difficult in distinguishing the patterns that require joint spatial and graph consideration. 2) Difficult in jointly maintaining that the representation is invariant to node permutations, rotation and translation transformations. 3) High efficiency and scalability according to the requirement of handling incremental information.

## 4 Approach

In order to achieve the novel spatial network representation learning by addressing the above-mentioned challenges, we propose a new method named spatial graph message passing neural network (SGMP) and a new accelerating algorithm which relies on sampling random spanning trees. Specifically, to discriminate spatial networks especially for the spatial-graph joint patterns, we propose a new message passing scenario which aggregates the node spatial information via higher-order edges as shown in Figure 3(a) and elaborated in Section 4.2. This scenario preserves graph and spatial information while aggregation with theoretical guarantees. To ensure that the representation is invariant to rotation and translation transformations, we propose to characterize several geometric properties in *length three path*, which is proved to represent node spatial information with guarantee on the properties of *rotation-invariant*, *translation-invariant*, and *information-lossless*. This is illustrated in Figure 3(b) and will be detailed in Section 4.1. To address the efficiency issue, an innovative sampling algorithm for accelerating training named Kirchhoff-normalized graph-sampled random spanning tree is proposed. The algorithm reduces the time and space complexity from $O(N^3)$ to $O(N)$ while still stay equivalent to original graph, which will be discussed in details in Section 4.3.

### 4.1 Node Spatial Information Representation

As mentioned above and in Figure 2, we need a novel way to represent the node spatial information that can preserve all the spatial structure information losslessly and also maintain rotation and translation invariance. We cannot directly use the Cartesian coordinates because they are not rotation- and translation-invariant. Although there are conventional node spatial information representation methods that maintained the rotation and translation invariance in the domain of spatial deep learning [78, 36], we cannot simply use them to handle spatial networks because they cannot consider the confinement on neighborhood from graph perspective. Otherwise, the coupled spatial-graph properties cannot be captured. Therefore, we consider to leverage *length $n$ path* to represent the node spatial information. The most simplest way is to just use the distance among nodes and we can have $n = 4$ to ensure the spatial information is preserved. However, we want to minimize the length of the path since the size of the neighborhood grows with a factor of $O(N)$ when one more length for the path is considered. To achieve this, we successfully reduce $n$ to 3 by proposing a new spatial information representation on path, where we use geometry features *distance*, *angle*, and *torsion* as detailed in the following equation and also illustrated in Figure 3(b).

The spatial information of a spatial network $S = (G, P)$ with $N$ nodes can be expressed as a set of Cartesian coordinates $P = \{(x_i, y_i, z_i) | x_i, y_i, z_i \in \mathbb{R}\}_{i=1}^N$. It can also be represented as $\mathbf{P} \in \mathbb{R}^{N \times 3}$ in a matrix form. The set of all *length $n$ path* starts from node $v_i$ can be represented as $\Pi_n^i$. Particularly, a *length three path* $v_i \to v_j \to v_k \to v_p$ can be expressed as $\pi_{ijkp} \in \Pi_3^i$. Given a spatial network $S$ where its graph $G$ is strongly connected and the longest path $\zeta \geq 3$, the proposed

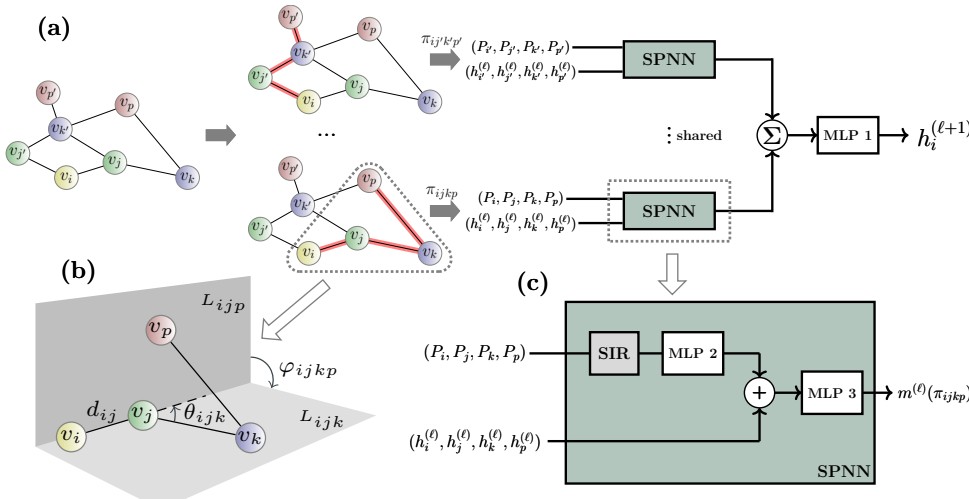

Figure 3: Illustration of the proposed spatial graph message passing neural network (SGMP). (a) The process of updating the hidden state embedding $h_i^{(\ell)}$ of node $v_i$ by aggregating the spatial-graph message information from *length three path*. (b) An example to illustrate each elements in our spatial information representation (Equation 1). Here $L_{ijp}$ is the plane defined by node $v_i$, $v_j$ and $v_p$ and $L_{ijk}$ is the plane defined by node $v_i$, $v_j$ and $v_k$. (c) This is the spatial path neural network block which is designed to learn the coupled spatial-graph property. This block also maintains the invariance to rotation and translation transformations by the spatial information representation (SIR).

spatial information representation can be expressed by one of its *length three path* $\pi_{ijkp} \in \Pi_3^i$ as

$$(d_{ij}, d_{jk}, d_{jp}, \theta_{ijk}, \theta_{ijp}, \varphi_{ijkp}), \tag{1}$$

where

$$
\begin{aligned}
&d_{ij} = ||\mathbf{P}_{ij}||_2, d_{jk} = ||\mathbf{P}_{jk}||_2, d_{jp} = ||\mathbf{P}_{jp}||_2, \\
&\theta_{ijk} = \arccos(\langle \frac{\mathbf{P}_{ij}}{d_{ij}}, \frac{\mathbf{P}_{jk}}{d_{jk}} \rangle), \theta_{ijp} = \arccos(\langle \frac{\mathbf{P}_{ij}}{d_{ij}}, \frac{\mathbf{P}_{jp}}{d_{jp}} \rangle), \\
&\varphi_{ijkp} = \text{Parity} \cdot \bar{\varphi}_{ijkp}, \\
&\mathbf{n_{ijk}} = \frac{\mathbf{P}_{ij} \times \mathbf{P}_{jk}}{||\mathbf{P}_{ij} \times \mathbf{P}_{jk}||_2}, \mathbf{n_{ijp}} = \frac{\mathbf{P}_{ij} \times \mathbf{P}_{jp}}{||\mathbf{P}_{ij} \times \mathbf{P}_{jp}||_2}, \\
&\bar{\varphi}_{ijkp} = \arccos(\langle \mathbf{n_{ijk}}, \mathbf{n_{jkp}} \rangle), \\
&\text{Parity} = \langle \frac{\mathbf{n_{ijk}} \times \mathbf{n_{ijp}}}{||\mathbf{n_{ijk}} \times \mathbf{n_{ijp}}||_2}, \frac{\mathbf{P}_{ij}}{||\mathbf{P}_{ij}||_2} \rangle.
\end{aligned}
\tag{2}
$$

**Theorem 1.** *Here the distances $d_{ij} \in [0, \infty)$, angles $\theta_{ijk} \in [0, \pi)$ and torsions $\varphi_{ijkp} \in [-\pi, \pi)$ are rigorously invariant under all rotation and translation transformations $\mathcal{T} \in \text{SE}(3)$.*

The proof for Theorem 1 is straightforward and can be found in Appendix B. It is remarkable to mention that the proposed representation in Equation 1 not only satisfies the invariance under rotation and translation transformation but also retains the necessary information to reconstruct the original spatial networks under weak conditions, as described in the following theorem.

**Theorem 2.** *Given a spatial network $S = (G, P)$, if $G$ is a strongly connected graph with longest path $\zeta \geq 3$, then given Cartesian coordinates of three non-collinear connected nodes $(v_j, v_k, v_p)$ in a length three path $\pi_{ijkp}$ of one node $v_i$, the Cartesian coordinates $P$ can be determined by the representation defined in Equation 1.*

The proof to this theorem is a consequence of the following lemma, which is proved in Appendix B.

**Lemma 1.** *Given Cartesian coordinates of three non-collinear connected nodes $(v_j, v_k, v_p)$ in a length three path $\pi_{ijkp}$ of one node $v_i$, the Cartesian coordinate $P_i$ of node $v_i$ can be determined by the representation defined in Equation 1.*

Now we can prove Theorem 2. As stated in Lemma 1, the Cartesian coordinate of node $v_i$ can be determined by its connected neighbors $v_j, v_k, v_p$ in the path of $\pi_{ijkp}$. Due to the property of strong connectivity of graph $G = (V, E)$, we can repeatly solve the coordinate of a connected node to the set of nodes with known coordinates. Thus, start from an arbitrary *length three path* the Cartesian coordinates $P$ of whole spatial networks is determined.

## 4.2 Spatial Graph Message Passing Neural Network (SGMP)

Spatial network representation learning requires us to do convolution that aggregates jointly the graph and spatial information from the graph neighborhood. The most important issue is to maintain the discriminative power without loss of graph and spatial information during the aggregation operation. In the meanwhile, we need to maintain permutation-invariant, rotation- and translation-invariant. To achieve this, we propose the following operation to update the hidden state embedding $h_i^{(\ell)}$ of node $v_i$ by aggregate the messages passing on all its *length three path* $\Pi_3^i$:

$$h_i^{(\ell+1)} = \sigma^{(\ell)}\Big(\text{SUM}\big(\{m^{(\ell)}(\pi_{ijkp})|\pi_{ijkp} \in \Pi_3^i\}\big)\Big), \tag{3}$$

where $\sigma^{(\ell)}$ is a multilayer perceptron (MLP) with ReLU as activation function and the spatial-graph interacted message $m^{(\ell)}(\pi_{ijkp})$ is generated by a spatial path neural network (SPNN) block:

$$\begin{aligned} m^{(\ell)}(\pi_{ijkp}) &= \phi^{(\ell)}\Big(\bar{m}^{(\ell)}(\pi_{ijkp}), \psi^{(\ell)}\big(\hat{m}(\pi_{ijkp})\big)\Big), \\ \bar{m}^{(\ell)}(\pi_{ijkp}) &= (h_i^{(\ell)}, h_j^{(\ell)}, h_k^{(\ell)}, h_p^{(\ell)}), \\ \hat{m}(\pi_{ijkp}) &= (d_{ij}, d_{jk}, d_{jp}, \theta_{ijk}, \theta_{ijp}, \varphi_{ijkp}), \end{aligned} \tag{4}$$

where $\phi^{(\ell)}$ and $\psi^{(\ell)}$ are two nonlinear functions to extract the complicated coupling relationship between spatial and graph information, in which we use the multilayer perceptron (MLP) with ReLU as the activation function in our settings.

Finally, the representation of spatial network $S$ can be achieved by applying a graph aggregation operation: $f(S) = \text{AGG}(\{h_i^{(K)}|v_i \in G\})$, where AGG is a permutation invariant function such as SUM or MEAN, and $K$ is the number of our message passing operation layers.

Since the node spatial information is already rotation- and translation-invariant, these properties can be intrinsically preserved by the operation in Equation 3. Node permutation will also be preserved due to the usage of the permutation invariant function SUM. Moreover, the following theorem proves that the discriminative power is also preserved from the perspective of maintaining the necessary spatial information, when the dimensions of hidden state embedding are sufficiently large.

**Theorem 3.** *Let $\mathcal{S}$ denote the collection of spatial networks with $N$ nodes given the graph $G = (V, E)$, and $\mathcal{F}$ denote the class of our SGMP functions while $\gamma$ is a continuous function. Suppose $g : \mathcal{S} \to \mathbb{R}$ is a continuous set function. For all $\epsilon > 0$, there exists a function $f \in \mathcal{F}$, such that for any $S \in \mathcal{S}$,*

$$|g(S) - \gamma(f(S))| < \epsilon. \tag{5}$$

The proof can be found in Appendix B. The key idea to the proof of this theorem is that we can discretize the continuous spatial information by partitioning the space into voxels.

## 4.3 Accelerate Training through Sampling Random Spanning Trees

Note that our model is a high order message passing neural network whose time and memory consumption is cubic to the average number of node degree. To reduce the complexity of graph neural networks, a typical way is based on sampling [96]. Many graph-sampling methods have been proposed for accelerating graph neural network [45, 16], which typically focus on randomly extracting a subgraph from the original graph. However, they cannot guarantee the generated graph is a strongly connected graph, which is required by our node spatial information representation in order to maintain no information loss. To ensure that the sampled graphs are connected and sparse, we innovatively propose a Kirchhoff-normalized graph-sampled random spanning tree method for accelerating the training. The proposed method largely reduces the complexity and maintains the equivalence to the original graph. Specifically, a spanning tree $T = (V, E_T)$ of an undirected graph $G = (V, E)$ that is a tree which contains all vertices in $G$. The number of edges of spanning trees is

$|E_T| = |V| - 1$, which implies that the time and space complexity during training will not be affected by the number of original edges $|E|$ in graph $G$. We modify our updating operation in Equation 1 as

$$h_i^{(\ell+1)} = \sigma^{(\ell)}\Big(\text{SUM}\big(\{m^{(\ell)}(\pi_{ijkp}) | \pi_{ijkp} \in \bar{\Pi}_{T,3}^i\}\big)\Big), \tag{6}$$

where we use $\bar{\Pi}_{T,3}^i$ denotes the set of all *length three path* starts from node $v_i$ in a sampled spanning tree $T = (V, E_T)$. It is noticed in Equation 6 that randomly sampling spanning trees $T$ from the original graph $G$ will introduce an uneven probability distribution for edges, which results in non-uniform weights for path messages in our proposed message passing layer. Here we introduce the Kirchhoff-normalized method to remove the uneven distribution by pre-computing the sampling probability of a path $\pi_{ijkp}$ in a sampled random spanning tree $T$. We further modify the Equation 6 as

$$h_i^{(\ell+1)} = \sigma^{(\ell)}\Big(\text{SUM}\big(\{\frac{m^{(\ell)}(\pi_{ijkp})}{q(\pi_{ijkp})} | \pi_{ijkp} \in \bar{\Pi}_{T,3}^i\}\big)\Big), \tag{7}$$

where $q(\pi_{ijkp})$ is the sampled probability of path $\pi_{ijkp}$ in a random spanning tree.

**Proposition 1.** *Let $T$ denote a uniformly random spanning tree of a graph G. Then for a length three path $\pi_{ijkp} = (e_{ij}, e_{jk}, e_{kp})$ we have that*

$$\Pr(\pi_{ijkp} \in T) = det[Y_{\pi_{ijkp}}], \tag{8}$$

where $Y$ is called the *transfer function matrix* [9]. The proof is achieved by applying graph theory theorems including Kirchhoff matrix tree theorem [14] and Burton-Pemantle theorem [12], which can be found in Appendix B. The following result establishes that the approximated form in Equation 7 is consistent to original form.

**Proposition 2.** *If $\sigma^{(\ell)}$ is continuous, the expectation of the approximated form in Equation 7 converges surely to the original form in Equation 3 when the number of samples is sufficiently large.*

The proof is a consequence of the strong law of large numbers and the continuous mapping theorem, which can also be found in Appendix B.

**Complexity analysis of a single layer.** Consider a spatial network with $N$ nodes and dense edge data, our full SGMP layer has $O(N^3)$ time and space complexity according to the size of the neighborhood. Our accelerating algorithm based on sampling random spanning tree, however, has only $O(N)$ time and space complexity as only $N - 1$ edges exist in the generated spanning trees.

## 5 Experiments

In this section, the experimental settings are introduced first, then the performance of the proposed method is presented through a set of comprehensive experiments. All experiments are conducted on a 64-bit machine with an NVIDIA GPU (GTX 1080 Ti, 11016 MHz, 11 GB GDDR5). The proposed method is implemented with Pytorch deep learning framework [70].

### 5.1 Experiment Setup

**Datasets** • *(i) Synthetic dataset.* The spatial growth graph model [6] is a spatial variant of the preferential attachment model proposed by Albert and Barabasi [1], which describes that spatial information concerns the formation of networks and long-range links are usually connecting the hubs (well-connected nodes). The process to generate such spatial networks starts from an initial connected network of $m_0$ nodes and introduces a new node $n$ at each time step. The new node is allowed to make $m \le m_0$ connections towards existing nodes with a probability $\Pi_{n \to i} \sim k_i F[d_E(n, i)]$, where $k_i$ is the degree of node $i$ and $F$ is an exponential function $F(d) = e^{-d/r_c}$ of the euclidean distance $d_E(n, i)$ between the node $n$ and the node $i$ [5]. General characteristics of spatial networks [6] such as clustering coefficient $\mu$, spatial diameter $D$, spatial radius $r$ are set as the prediction targets. Besides, we also add the interaction range $r_c$, which is a significant coupled spatial-graph label that affects the formation of the spatial networks, as another prediction target. We vary the size and other parameters (according to Appendix. C for details) of spatial networks to collect $3,200$ samples in our synthetic dataset. • *(ii) Real-world molecular property datasets.* We experiment on 5 chemical molecule benchmark datasets from [94], including both classification (BACE, BBBP) and regression (ESOL, LIPO, QM9). Particularly, QM9 is a multi-task regression benchmark with 12 quantum mechanics

| Taret | $\mu$ | $D$ | $r$ | $r_c$ |
|---|---|---|---|---|
| GIN | 0.136(.007) | 1.015(.047) | 0.659(.029) | 1.616(.075) |
| GAT | 0.129(.001) | 1.291(.049) | 0.888(.014) | 1.716(.017) |
| GatedGNN | 0.089(.013) | 0.753(.074) | 0.481(.066) | 1.411(.031) |
| PointNet | 0.129(.003) | 0.912(.030) | 0.615(.020) | 1.551(.066) |
| PPFNet | 0.106(.006) | 0.747(.037) | 0.527(.014) | 1.377(.057) |
| SGCN | 0.133(.003) | 1.269(.055) | 0.856(.044) | 1.736(.020) |
| SchNet | 0.128(.001) | 1.006(.058) | 0.686(.031) | 1.691(.039) |
| DimeNet | 0.103(.027) | 1.266(.147) | 0.556(.094) | 1.412(.059) |
| SGMP | **0.068(.005)** | 0.748(.168) | 0.450(.046) | 1.332(.031) |
| SGMP (with st) | 0.088(.001) | **0.291(.021)** | **0.252(.023)** | **1.266(.019)** |

Table 1: Root mean square error (RMSE) results on synthetic dataset. Here $\mu$ is clustering coefficient, $D$ is spatial diameter, $r$ is spatial radius and $r_c$ is the interaction radius in the formation of spatial growth graph.

| Task | Regression | | | Classification | |
|---|---|---|---|---|---|
| Dataset | ESOL | LIPO | HCP | BACE | BBBP |
| GIN | 0.776(.021) | 0.699(.047) | 0.792(.133) | 0.792(.025) | 0.864(.020) |
| GAT | 0.783(.053) | 0.757(.049) | 0.561(.037) | 0.780(.035) | 0.854(.025) |
| GatedGNN | 0.675(.050) | **0.630(.034)** | 0.566(.036) | 0.816(.023) | 0.858(.020) |
| PointNet | 0.716(.036) | 0.708(.030) | 0.720(.123) | 0.799(.023) | 0.843(.027) |
| PPFNet | 0.731(.054) | 0.720(.037) | 0.680(.065) | 0.805(.032) | 0.869(.023) |
| SGCN | 0.743(.056) | 0.726(.055) | 0.674(.059) | 0.778(.030) | 0.849(.021) |
| SchNet | 0.697(.051) | 0.691(.058) | 0.593(.037) | 0.803(.032) | 0.864(.036) |
| DimeNet | 0.730(.047) | 0.666(.047) | 0.818(.127) | 0.791(.031) | 0.864(.036) |
| SGMP | 0.646(.049) | 0.695(.027) | **0.524(.046)** | **0.830(.021)** | **0.880(.020)** |
| SGMP (with st) | **0.612(.054)** | 0.699(.021) | 0.555(.045) | 0.811(.024) | 0.873(.024) |

Table 2: Results for four molecule property datasets and the HCP brain network. We report accuracy score for BACE and BBBP datasets, root mean square error (RMSE) for ESOL and LIPO, and mean average error (MAE) for HCP brain network dataset.

properties. The data is obtained from the pytorch-geometric library [35]. ● *(iii) Real-world HCP brain network dataset.* We also conducted an experiment using the structural connectivity (SC) of the brain network to predict the age of the subjects, which is a significant task in understanding the aging process of the human brain [50]. In specific, SC is processed from the Magnetic Resonance Imaging (MRI) data obtained from the human connectome project (HCP) [82]. By following the preprocessing procedure in [84], the SC data is constructed by applying probabilistic tracking on the diffusion MRI data using the Probtrackx tool from FMRIB Software Library [49] with 68 predefined regions of interest (ROIs). Then a threshold is applied to SC data to construct the brain networks [77, 37]. The spatial coordinates of regions are expressed as the center point of each region.

**Comparison methods.** To the best of our knowledge, there has been little previous work to handle the generic spatial networks. Spatial graph convolutional networks (SGCN) is a recently proposed method to handle generic spatial networks by applying a convolution operation to learn the spatial-graph interacted information using the relative coordinates between nodes and their first-order neighbors. In addition, we compare with three strong graph neural networks (GIN, GAT and Gated GNN) methods and four spatial neural networks (PointNet, PPFNet, SchNet, and DimeNet) methods for comparisons. For methods in the class of GNNs, we feed the Cartesian coordinates as node attributes while we add the node attribute and graph connectivity information to the class of SNNs for a fair comparison. The details about the benchmark models can be found in Appendix. C. Besides the models above, we also compare our model with a state-of-the-art higher-order graph neural networks PPGN [63]) in the QM9 benchmark, the results are provided from original authors.

### 5.2 Experimental Performance

In this section, the performance of the proposed method and its accelerated algorithm with sampling random spanning tree (with st), as well as other methods on both synthetic and real-world datasets are presented first. Then we present the efficiency test on our sampling random spanning trees method. In addition, we measure the exactness of invariance of our proposed model under translation and rotation transformations.

**Effectiveness results.** ● *(i) Synthetic Dataset.* Table 1 summarizes the effectiveness comparison for the synthetic dataset, where our proposed SGMP model with sampling spanning tree outperforms the best benchmark model (GatedGNN) by $35.7\%$ on average. Especially, our model achieves lower error on the target of interaction radius ($r_c$), which proves that our proposed model can better capture and exploit the significant coupled spatial-graph characteristics in spatial networks. ● *(ii) Real-world Datasets.* Table 2 presents the results of four molecule property datasets and the HCP brain network dataset, where our proposed method achieves the best results in 4 out of 5 datasets. The results for the QM9 dataset are presented in Table 3, where our proposed method demonstrates its strength through outperforming the benchmark methods in 10 out of 12 targets, which is an improvement by over $14\%$ on average. Particularly, we notice that the performance of the class of SNNs achieved significantly better results than the class of GNNs by a $38\%$ improvement on average, which arguably implies that the quantum mechanics targets of the QM9 dataset are dominated by the spatial information. In addition, the group of jointly-spatial-graph-based methods achieved a $68.9\%$ improvement compared to the group of point-cloud-based methods. The twelve quantum mechanical properties in the QM9 dataset seems highly related to the spatial geometry properties between nodes. For example, the formation energy ($U$) is related to the distances, angles, and torsions among nodes. In this situation, we notice that the performance of the group of point-cloud-based methods is significantly better than the group of GNN based methods, and jointly-spatial-graph-based methods can better explore the coupled spatial-graph property.

| Target | GIN | GAT | Gated | PointNet | PPFNet | SGCN | PPGN | SchNet | DimeNet | SGMP | SGMP (with st) |
|---|---|---|---|---|---|---|---|---|---|---|---|
| $\mu$ | 0.583 | 0.661 | 0.543 | 0.465 | 0.503 | 0.503 | **0.093** | 0.452 | 0.360 | 0.130 | 0.187 |
| $\alpha$ | 0.652 | 0.952 | 0.609 | 0.453 | 0.459 | 0.531 | 0.318 | 0.347 | 0.189 | **0.113** | 0.174 |
| $\epsilon_{\text{HOMO}}$ | 269.5 | 326.7 | 206.2 | 158.6 | 151.9 | 193.8 | 47.3 | 347.4 | 78.6 | 64.7 | **45.7** |
| $\epsilon_{\text{LOMO}}$ | 175.4 | 237.1 | 135.4 | 123.8 | 136.9 | 141.7 | 57.1 | 151.6 | 61.0 | **44.7** | 67.9 |
| $\delta_\epsilon$ | 361.4 | 510.3 | 314.4 | 245.5 | 221.9 | 275.5 | **78.9** | 120.6 | 103.7 | 83.7 | 98.8 |
| $\langle R^2 \rangle$ | 63.7 | 97.1 | 63.1 | 34.5 | 27.8 | 34.9 | 3.8 | 213.2 | 14.13 | 5.9 | **3.6** |
| ZPVE | 12.3 | 15.7 | 12.0 | 7.0 | 7.4 | 7.4 | 10.8 | 34.3 | 3.1 | 2.3 | **2.0** |
| $U_0$ | 260.1 | 335.9 | 222.5 | 112.7 | 153.5 | 201.3 | 36.8 | 101.7 | 26.8 | **26.1** | 31.9 |
| $U$ | 262.9 | 326.1 | 244.7 | 115.5 | 160.5 | 210.1 | 36.8 | 107.5 | 27.8 | **25.2** | 34.8 |
| $H$ | 269.0 | 329.7 | 239.2 | 123.1 | 157.6 | 199.2 | 36.8 | 107.0 | 27.9 | **27.5** | 31.3 |
| $G$ | 252.7 | 314.1 | 221.1 | 124.3 | 158.4 | 207.8 | 36.4 | 95.0 | 25.8 | **24.6** | 28.2 |
| $c_V$ | 0.344 | 0.430 | 0.283 | 0.196 | 0.221 | 0.277 | 0.055 | 0.452 | 0.064 | **0.043** | 0.064 |

Table 3: The mean average error (MAE) results for QM9 dataset.

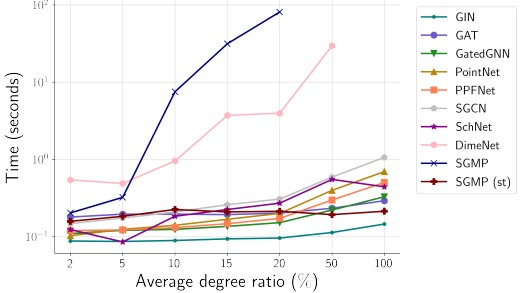

Figure 4: Efficiency analysis of our proposed models and all benchmark models. Note that our proposed algorithm with sampling random spanning tree significantly improves the scalability and efficiency.

| ADR (%) | wo st | w st | speed up |
|---|---|---|---|
| 2 | 0.203s | 0.158s | 1.3× |
| 5 | 0.323s | 0.184s | 1.7× |
| 10 | 7.52s | 0.225s | 33.4× |
| 15 | 31.43s | 0.209s | 150.3× |
| 20 | 80.96s | 0.213s | 379.7× |
| 50 | - | 0.193s | - |
| 100 | - | 0.214s | - |

Table 4: Training time per epoch for our full model without sampling spanning tree (wo st) and accelerating method with sampling spanning tree (w st). (-) indicates an out-of-memory error. The sampling algorithm is on average 113 times faster than our full method. ADR is short for the average degree ratio.

**Efficiency analysis.** To validate the efficiency of the proposed sampling random spanning tree algorithm, we use our HCP brain network dataset with different thresholds on structural connectivity (SC) to obtain different average degrees for the nodes. The number of nodes is a fixed number (68) while we vary the average of degrees ratio (ADR= $\frac{E}{E_f}$, where $E$ is the number of edges and $E_f$ is the number of edges in complete graphs, e.g. ADR= $100\%$ indicates a complete graph). We report the results of the average training time per epoch among all models for 20 epochs. As shown in Figure 4 and Table 4, our accelerating algorithm achieves significant improvements in training efficiency. Note that our method is even faster than most of the first-order methods when the graph connections are dense (ADR over $50\%$). Notice that higher-order methods (e.g. our full method is third-order and DimeNet is second-order) are unable to handle complete graphs due to the limits of GPU memory. The scalability of our sampling method is remarkable, which can maintain a constant time and space complexity with the increasing number of connected edges.

**Rotation and translation invariant test.** Similar to previous work [36], we also measure the rotation and translation robustness by uniformly adding translation and rotation transformations to the input Cartesian coordinates. Here we only report the accuracy results of classification task on the molecular dataset BACE due to the space limit while the results are similar on all datasets. According to Figure 5, we can note that the performance of our proposed model stays invariant under both translation and rotation transformations. SchNet and DimeNet can also achieve invariance under transformations because they also only use the rotation- and translation-invariant spatial features in their models. PPFNet can stay invariant under rotation transformations but not translation transformations because it preserves the origin in the model. On the other hand, SGCN can stay invariant under translation transformations but not rotation invariant because it only utilizes relative coordinates. This experiment validates the importance of applying a rotation- and translation-invariant model since we can observe that the performance of models without a theoretical guarantee drop significantly under adding rotation and translation transformations.

## 6   Conclusion

This paper focuses on the crucial problem of learning powerful representations from spatial networks, which has tightly coupled spatial and graph information that can not be addressed by applying spatial and network methods separately. The proposed spatial graph message passing neural network (SGMP) effectively addresses the unique challenges in spatial networks by jointly considering the spatial and graph properties, and still maintain the invariance to node permutations, as well as rotation and

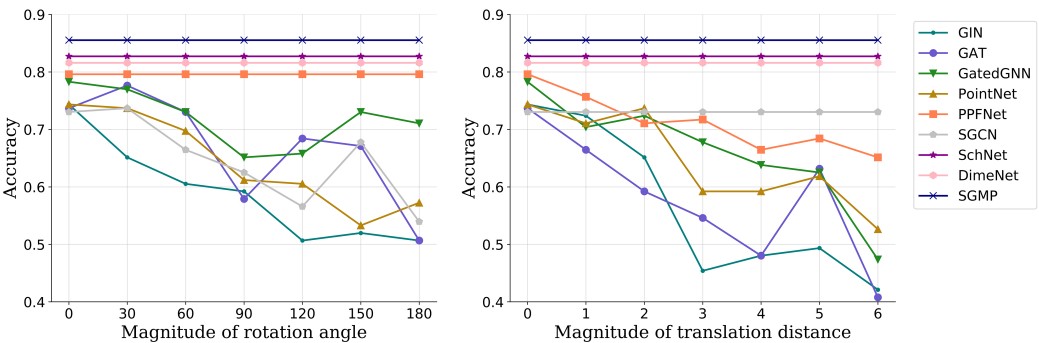

Figure 5: Robustness test of rotation and translation invariant: $x$-axis shows data augmentation on the test set. The $x$-value corresponds to the magnitude of rotation angle (left) or translation distance (right). The $y$-axis shows the accuracy score on the test set.

translation transformations. In addition, our proposed accelerating algorithm largely alleviates the efficiency issue in solving spatial network issues. Experimental results on synthetic and real-world datasets demonstrate the outstanding discriminative power of our model, and the efficiency test shows a remarkable improvement in training time and scalability of our proposed accelerating method.

## Acknowledgement

This work was supported by the National Science Foundation (NSF) Grant No. 1755850, No. 1841520, No. 2007716, No. 2007976, No. 1942594, No. 1907805, a Jeffress Memorial Trust Award, Amazon Research Award, NVIDIA GPU Grant, and Design Knowledge Company (subcontract number: 10827.002.120.04). The authors acknowledge Emory Computer Science department for providing computational resources and technical support that have contributed to the experimental results reported within this paper.

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
