# OpenReview forum: "Representation Learning on Spatial Networks"
_NeurIPS.cc/2021/Conference — NeurIPS 2021 Poster_

### Official Review · Reviewer_wGJF · 2021-07-14

**Rating:** 8
**Confidence:** 4

**Summary:**

This paper advances graph representation learning into spatial networks, which is closer to the networks in physical world that are embedded in (3D) space and impacted by geometric perspectives. This work demonstrated how existing GNN or point cloud methods are not sufficient and then propose their new geometric-driven graph convolution layers which preserves discriminative power, rotation- and translation-invariant. The authors also propose a graph sampling method which ensure graph-connectivity, sparsity, and recoverability. Experimental results on 5 real-world benchmark datasets and one synthetic dataset demonstrate the effectiveness, efficiency, and geometric-property preservation over several comparison methods.

**Limitations And Societal Impact:**

There is no obvious limitation or potential negative societal impact of this work.

**Main Review:**

This paper focuses on spatial network representation learning problem. Spatial networks encompass both graph information and geometric information such as node coordinates. This paper first rigorously analyzes why spatial graph pattern representation learning requires a coupled consideration on geometric properties and graph isomorphism, and why it cannot be handled by the existing works on merely graphs or point clouds individually. Then the authors propose new spatial network convolution layers which considers higher-order graph neighborhood in order to preserve geometric shape properties. Theoretical analyses/guarantees have been provided for the properties such as rotation-invariant, translate-invariant, and discriminative power.

To handle the computational complexity issue caused by the higher-order neighborhood consideration, the authors propose an acceleration technique by sampling min-spanning trees, which are sparsest graphs that maintain graph connectivity. This way the authors claim that it can reduce the computational complexity to almost linear, even it is higher-order convolution. The authors also give theorems on the effectiveness of the above approximation. Experiments on five real-world benchmark datasets and synthetic datasets are conducted where the proposed method have, for example, outperformed the state-of-the-art methods by >=30% on synthetic datasets and >=15% on QM9 dataset. Ablation studies have demonstrated the efficiency of the proposed spanning tree sampling methods, which provide a speed-up up to 300 times with similar prediction accuracy. Rotation- and translate-invariant properties of the learned representations have also been examined and seems apparently outperformed existing work.

Overall, I like this paper which seems solid in both theoretical aspect and experimental perspective. However, there are still several (minor) issues, as follows:

- Need to give more explanations on why spatial information must be represented in Equation (1). Many terms in Equation (2) have no sufficient explanations. I understand in Theorem 1 the authors have proved Equation (1) can provide rotation- and translation- invariant properties. But is Equation (1) the simplest way to formalize it?
- Some experiment results require more explanations. Despite the good performance shown in Tables 2 and 3, the authors need to provide more detailed analyses on the results. For example, in order to see the potential advantage/disadvantage, analysis on the comparisons is needed among purely-GNN-based methods, purely-point-cloud based methods, and jointly-spatial-graph-based methods.


**Time Spent Reviewing:**

6

---

> ### Author Response · Authors · 2021-08-10
> **Response to Reviewer wGJF**
>
> Thank you for the detailed views and insightful feedback!
>
> Q1. "Need to give more explanations on why spatial information must be represented in Equation (1)."
>
> A1. The proposed representation in Equation (1) not only satisfies rotation- and translation-invariant but also retains the necessary information of the geometrical structure of spatial networks. Jointly using distances, angles, and torsions, which are commonly used geometry features in the three-dimensional geometry study area [1], can ensure information-lossless. Specifically, distance and angle can locate nodes to a circle, with using torsion, the intersection ensures the location is pinpointed. The more detailed proof is shown in Theorem 2.
> [1] Thurston, William P. Three-dimensional geometry and topology. Vol. 1. Princeton university press, 1997.
>
> Q2. "in order to see the potential advantage/disadvantage, analysis on the comparisons is needed among purely-GNN-based methods, purely-point-cloud based methods, and jointly-spatial-graph-based methods"
>
> A2. In lines 337-339, we have a discussion on the comparison between the group of GNN methods and the group of point-cloud-based methods. We notice that the performance of the group of point-cloud-based methods achieved significantly better results than the group of GNNs by a 38% improvement on average. In addition, the group of jointly-spatial-graph-based methods achieved a 68.9% improvement compared to the group of point-cloud-based methods. The twelve quantum mechanical properties in the QM9 dataset seems highly related to the spatial geometry properties between nodes. For example, the formation energy ($U$) is related to the distances, angles, and torsions among nodes. In this situation, we notice that the performance of the group of point-cloud-based methods is significantly better than the group of GNN based methods, and jointly-spatial-graph-based methods can better explore the coupled spatial-graph property.

---

### Official Review · Reviewer_pPBn · 2021-07-16

**Rating:** 6
**Confidence:** 3

**Summary:**

The paper proposed a new graph representation learning method specifically for spatial networks. This method considered both spatial and network information with a theoretical guarantee with node permutation, rotation and translation invariance by introducing length three path. Additionally, an approach was developed to improve time and memory efficiency. Its ability of network representation has been demonstrated in multiple synthetic and real-world datasets. Overall, this paper is easy to follow and well-written.

**Limitations And Societal Impact:**

I don't see any potential negative societal impact

**Main Review:**

The method is new and easy to understand. Their main contributions lie in three aspects. First, use geometry features (i.e., distance, angle, and torsion) to represent spatial information. These good geometry features remain invariant under rotation and translation transformation. Second, based on geometry features as well coordinate representations of points, they proposed Spatial Graph Message Passing Neural Network. Third, an algorithm to speed up the training process.

Questions: 1) As said, Cartesian coordinates are not rotation- and translation-invariant. But why still remain them in Eq.4 when geometry features are used to represent spatial information and incorporated. It would be interesting to see the impacts of coordinates of points vs. geometry properties individually on the performance. 2) Essentially, SGMPNN is a path-based method rather than classic graph representation methods, where local graph structures are used in aggregation. 3) It is not very clear explained how authors decided on those three geometry features.

The tasks are not new and even questionable (e.g., why it is useful to predict clustering efficient), although they are conventional and the method was proved to perform well on these tasks. I am wondering to which practical applications the method can be applied.

Their results are sound. In Table 1, 2 and 3, the description of SGMP (st) was not given, though readers may infer st refers to Spanning Tree. Besides, SGMP (with st) and SGMP performed very differently shown in those tables. Authors may explain why one is better than the other in some cases while is beaten in other cases.

**Time Spent Reviewing:**

4

---

> ### Author Response · Authors · 2021-08-10
> **Response to Reviewer pPBn**
>
> We thank the reviewer for their insightful reviews.
>
> Q1. "Cartesian coordinates are not rotation- and translation-invariant. But why still remain them in Eq.4 when geometry features are used to represent spatial information and incorporated. It would be interesting to see the impacts of coordinates of points vs. geometry properties individually on the performance."
>
> A1. There are no Cartesian coordinates in Eq. 4. In Eq. 4, the $h_{i}^{(\ell)}$ represents the hidden state embedding of node $v_i$ at layer $\ell$ while $\pi$ represents the used geometry features. Both $h_{i}^{(\ell)}$ and $\pi$ do not include Cartesian coordinates.
> Furthermore, thanks for your suggestions. We have added additional experiments to evaluate the impacts of coordinates of points vs. geometry properties. Specifically, we built a baseline method that uses the coordinates of points instead of the geometry properties in our original method. The classification accuracy results on the molecular dataset BACE are shown in the below table, where “coordinates” denotes the baseline while “geometry properties” denotes our method:
>
> --------------------------------------------------------------------------
>
>
> Rotation angle | coordinates | geometry properties
>
>    0 | 0.792 | 0.830
>
>    30 | 0.763 | 0.830
>
>    60 | 0.705 | 0.830
>
>    90 | 0.655 | 0.830
>
>
> --------------------------------------------------------------------------
>
> Our method using the geometry properties consistently outperformed the method using coordinates of points by 10.1% on average. In addition, the performance of the baseline method using coordinates of points drops significantly from 79.2% to 65.5% when additional rotation transformations are imposed to the data while the performance of our method using geometry properties stays invariant to rotation transformations.
>
> Q2. "Essentially, SGMPNN is a path-based method rather than classic graph representation methods, where local graph structures are used in aggregation."
>
> A2. Graph neural network methods are one of the mainstream methods for graph representation learning, where they define graph convolutions based on a node’s proximity relations.  Message passing neural networks is one type of graph neural network currently in use [1, 2]. Our proposed methods extend conventional message passing scenarios to spatial networks and aggregate the graph and spatial information jointly from the neighborhood in spatial networks.
>
> [1] Wu, Zonghan, et al. "A comprehensive survey on graph neural networks." IEEE transactions on neural networks and learning systems 32.1 (2020): 4-24.
>
> [2] Chen, Zhiqian, et al. "Bridging the gap between spatial and spectral domains: A survey on graph neural networks." arXiv preprint arXiv:2002.11867 (2020).
>
> Q3. "It is not very clear explained how authors decided on those three geometry features."
>
> A3. The proposed representation in Equation 1 (the geometry features) is motivated by the requirement of rotation-invariant, translation-invariance, and information-lossless properties with theoretical guarantee. The distance $d$, angle $\theta$, and torsion $\varphi$, which are commonly used geometry features in the three-dimensional geometry study area [1], is the data representation that can satisfy rotation- and translation-invariant. Jointly using them can ensure information-lossless. Specifically, distance and angle can locate nodes to a circle, with using torsion, the intersection ensures the location is pinpointed. The more detailed proof is shown in Theorem 2.
>
> [1] Thurston, William P. Three-dimensional geometry and topology. Vol. 1. Princeton university press, 1997.
>
> Q4. “The tasks are not new and even questionable (e.g., why it is useful to predict clustering efficient), although they are conventional and the method was proved to perform well on these tasks. I am wondering to which practical applications the method can be applied.”
>
> A4.
>
> 1. This paper has included several real-world datasets in computational chemistry and biological networks, which indicate the practicality of our method.
>
> 2. In addition to real-world datasets, we have also included a synthetic dataset, and clustering co-efficient is just for supporting a simple synthetic experiment that can validate method performance with high-quality labels. The design of such synthetic tasks follows benchmarks in classical studies on spatial networks [1].
>
> [1] Marc Barthélemy. Spatial networks. Physics Reports, 499(1-3):1–101, 2011.
>
> Q5. “SGMP (with st) and SGMP performed very differently shown in those tables. Authors may explain why one is better than the other in some cases while is beaten in other cases.”
>
> A5.
>
> 1. The performance of SGMP and SGMP (with st) is close in general: within 10% difference in 6 out of 7 datasets. In these 6 datasets, the difference is 4.1% on average.
>
> 2. The only relatively large difference happens in the prediction tasks of the spatial diameter $D$ and spatial radius $r$ on the synthetic dataset, where the performance of SGMP (with st) is even better than that of SGMP. This may be because SGMP (with st) is more advantageous for this specific task: the spatial diameter $D$ and spatial radius $r$ are defined by the spatial information instead of network topology. Therefore, since SGMP (with st) utilizes the network simplified by spanning tree, such simplification may potentially ease the learning task of our model towards competitive results compared to SGMP.

---

### Official Review · Reviewer_Jj9n · 2021-07-17

**Rating:** 6
**Confidence:** 4

**Summary:**

In this paper, the authors studied the problem of representation learning on spatial graphs. To achieve the goal, the authors proposed a generic framework by aggregating message passing mechanisms and spatial information. The authors also evaluated the proposed method on both synthetic data and real-world data.

**Limitations And Societal Impact:**

I suggest the authors introduce some potential real-world applications that the proposed method can be applied to, in a short paragraph, to show the potential social impact.

**Main Review:**

Strong Points:
1. The writing is clear and easy to follow.
2. The theoretical foundation is solid.


Concerns and Questions.
1. In Line 188, the authors claimed that "n=4 can ensure the spatial information is preserved". Why?

2, In Line 4, the authors claimed there exist very few works on the representation learning of spatial networks, which is not necessarily correct. In fact, there are some works studying representation learning for spatial graphs, or spatial graph embedding, which also considered spatial information while modeling topology, such as [1][2]
[1] Molecule property prediction based on spatial graph embedding
[2] Representation Learning for Spatial Graphs.
How does this work compare to the literature?

3. In the experiment, the authors compared the proposed method with GCN models. As the GCN models do not consider spatial information, I doubt whether it is fair to do the comparison.

**Time Spent Reviewing:**

1

---

> ### Author Response · Authors · 2021-08-10
> **Response to Reviewer Jj9n**
>
> We thank the reviewer for the insightful reviews.
>
> Q1. Elaboration on "n=4 can ensure the spatial information is preserved."
>
> A1. A strict proof for this claim is similar to Theorem 2 in our paper. We briefly describe the proof of this claim here: Given the Cartesian coordinates of four connected nodes $(v_j, v_k, v_p, v_q)$ that are not in a straight line, the Cartesian coordinates of node $v_i$ can be fully determined jointly by $d_{ij}$ , $d_{ik}$ , $d_{ip}$ , and $d_{iq}$ , namely the Euclidean distances between node $v_i$ and anyone among nodes $v_j$ , $v_k$ , $v_p$ , and $v_q$ . Thus, for any node in a spatial network, its coordinates can be determined in the above way, and hence the spatial structure information of the spatial network is preserved.
>
> Q2. "In fact, there are some works studying representation learning for spatial graphs, or spatial graph embedding, which also considered spatial information while modeling topology, such as [1][2]."
>
> A2. First, spatial networks are networks for which the nodes and edges are embedded in a geometric space, where the spatial and network properties are coupled together tightly. Previous works have at least one of the following limitations:
> 1.	Cannot simultaneously handle the spatial properties, network properties, and their interplay. Specifically, existing works typically purely consider spatial information (e.g. point cloud) and transfer it to graphs, and then perform techniques such as graph neural networks [1,3,4,7,8,9,10]. However, merely considering spatial information is not enough to handle typical spatial networks where their spatial information and network properties come from different and heterogeneous sources.
> 2.	Cannot guarantee important properties such as rotation invariance, translation invariance, and preserve all the geometrical structure information. Existing works typically utilize the off-the-shelf deep neural networks with Cartesian coordinates as inputs and a large amount of rotation-and translation-augmented data, which is computationally expensive and lacks theoretical guarantee of rotation- and translation-invariant on the representation [2,5]. Some existing works transform data into spherical representation and then apply spherical convolution operator to extract rotation- and translation- invariant features. Such transformation is deemed not a bijective mapping, so these methods inevitably suffer from the loss of geometric structure information in the learned spherical representation [3,4,6,10].
>
> Moreover, existing works usually are tailored for domain-specific applications, and cannot handle generic spatial networks. Existing works typically aim at domain-specific applications such as quantum chemistry [1,3,4,6,7,8,9] and computational materials [7,9], where they include domain-specific knowledge such as Schrodinger functions from quantum mechanics or density functional theory from computational materials.
>
> [1] Wang, Xiaofeng, et al. "Molecule property prediction based on spatial graph embedding." Journal of chemical information and modeling 59.9 (2019): 3817-3828.
>
> [2] Wang, Zheng, et al. "Representation Learning for Spatial Graphs." arXiv preprint arXiv:1812.06668 (2018).
>
> [3] Schütt, Kristof, et al. "SchNet: A continuous-filter convolutional neural network for modeling quantum interactions." NIPS. 2017.
>
> [4] Klicpera, Johannes, Janek Groß, and Stephan Günnemann. "Directional Message Passing for Molecular Graphs." International Conference on Learning Representations. 2019.
>
> [5] Danel, Tomasz, et al. "Spatial graph convolutional networks." International Conference on Neural Information Processing. Springer, Cham, 2020.
>
> [6] Gilmer, Justin, et al. "Neural message passing for quantum chemistry." International conference on machine learning. PMLR, 2017.
>
> [7] Unke, Oliver T., and Markus Meuwly. "PhysNet: a neural network for predicting energies, forces, dipole moments, and partial charges." Journal of chemical theory and computation 15.6 (2019): 3678-3693.
>
> [8] Anderson, Brandon, Truong Son Hy, and Risi Kondor. "Cormorant: Covariant Molecular Neural Networks." Advances in Neural Information Processing Systems 32 (2019): 14537-14546.
>
> [9] Chen, Chi, et al. "Graph networks as a universal machine learning framework for molecules and crystals." Chemistry of Materials 31.9 (2019): 3564-3572.
>
> [10] Esteves, Carlos, et al. "Learning so (3) equivariant representations with spherical cnns." Proceedings of the European Conference on Computer Vision (ECCV). 2018.
>
>
> Q3: "As the GCN models do not consider spatial information, I doubt whether it is fair to do the comparison."
>
> A3: The input into GCN models includes spatial information in our experiments. As mentioned in lines 318-320, we feed the Cartesian coordinates as node attributes for the method in the class of GNNs for a fair comparison.
>
> Q4. (Societal Impact) "I suggest the authors introduce some potential real-world applications that the proposed method can be applied to, in a short paragraph, to show the potential social impact."
>
> A4. Thank you for your suggestion. We have discussed the potential real-world applications in lines 28-31 that the proposed method can have a broad impact on many real-world applications, such as computational chemistry, biological networks, and human mobility analysis. We can further add some discussion about the potential societal impact on the perspectives of accelerating research discovery and building industry solutions for applications that involve spatial networks, such as drug design, mental disease diagnosis, and societal event synthesis in our later version.

---

> > ### Comment · Reviewer_Jj9n · 2021-08-12
> > **Potential Real-World Applications**
> >
> > Thank the authors for your clear response. Regarding real-world applications, human mobility modeling would be a promising direction. In fact, there are some works in this area using representation learning techniques, where geo-locations are considered as spatial graphs, such as [1][2][3][4][5]. But in these prior works, geometry features are not included. It would be better if the authors can consider discussing the connections of this paper to these prior works, and the potential adaption.
> >
> > [1] Region Representation Learning via Mobility Flow
> > [2] Incremental Mobile User Profiling: Reinforcement Learning with Spatial Knowledge Graph for Modeling Event Streams
> > [3] Unsupervised Representation Learning of Spatial Data via Multimodal Embedding
> > [4] Adversarial Substructured Representation Learning for Mobile User Profiling
> > [5] Exploiting Mutual Information for Substructure-aware Graph Representation Learning

---

> > > ### Author Response · Authors · 2021-08-16
> > > **Response to Questions related to Potential Real-World Applications**
> > >
> > > Q. “Regarding real-world applications, human mobility modeling would be a promising direction. In fact, there are some works in this area using representation learning techniques, where geo-locations are considered as spatial graphs, such as [1][2][3][4][5]. But in these prior works, geometry features are not included. It would be better if the authors can consider discussing the connections of this paper to these prior works, and the potential adaption.”
> > >
> > > A. Thank you very much for your valuable suggestion.
> > >
> > > 1. Comparison. Although some of these previous works used the term ‘spatial graph’ in their methods, they have a significant difference from the real definition of spatial graphs (e.g., the one defined in [6]). For example, the vertices of graphs in [2,4,5] are POI (Point of Interest) categories, hence such concept graphs are not physically embedded in a geometric space. Previous works in [1,3] only consider the distances between each pair of two contiguous nodes in each trajectory sequence but cannot guarantee to preserve a lot of key geometric information (e.g., the orientations, angles, and spatial distances between any nodes that are not neighbors). Different from these works, our method can preserve the 3D geometric information with a theoretical guarantee.
> > >
> > > 2. Connection. The main connection of our work to these prior works is that we all utilize the framework of graph neural networks to utilize the necessary network topology information. But our work effectively preserves the geometric information which can lead to better discriminative power of different mobility networks. Our work could also better consider higher-order neighborhood patterns in both network and spatial dimensions, which may be beneficial to capture richer human mobility patterns.
> > >
> > > 3. Adaption. Our work can be used to generate representations of human mobility networks with strong discriminative power. The learned representation vectors can be used for conventional downstream tasks on human mobility data. When the human mobility network resides in 2D space, our method can be readily adapted from 3D space to this situation by setting the z-coordinates to trivial a constant value (e.g., zeros).
> > >
> > > We would love to enrich the discussion on the human mobility networks in the paper with some points and references mentioned above.
> > >
> > > [1] Wang, Hongjian, and Zhenhui Li. "Region representation learning via mobility flow." Proceedings of the 2017 ACM on Conference on Information and Knowledge Management. 2017.
> > >
> > > [2] Wang, Pengyang, et al. "Incremental mobile user profiling: Reinforcement learning with spatial knowledge graph for modeling event streams." Proceedings of the 26th ACM SIGKDD International Conference on Knowledge Discovery & Data Mining. 2020.
> > >
> > > [3] Jenkins, Porter, et al. "Unsupervised representation learning of spatial data via multimodal embedding." Proceedings of the 28th ACM International Conference on Information and Knowledge Management. 2019.
> > >
> > > [4] Wang, Pengyang, et al. "Adversarial substructured representation learning for mobile user profiling." Proceedings of the 25th ACM SIGKDD International Conference on Knowledge Discovery & Data Mining. 2019.
> > >
> > > [5] Wang, Pengyang, et al. "Exploiting Mutual Information for Substructure-aware Graph Representation Learning." IJCAI. 2020.
> > >
> > > [6] Marc Barthélemy. Spatial networks. Physics Reports, 499(1-3):1–101, 2011.

---

### Decision · Program_Chairs · 2021-09-27

**Decision:**

Accept (Poster)

**Comment:**

The paper discusses the problem of representation learning for geometric graphs, which are graphs in which the vertices are positions in 3d space (or some other geometry), and the edges obey some local law.  These graphs come about in areas such as biology and chemistry.  Being able to compute certain functions on such objects, using deep networks, requires a good representation in latent space.  This paper suggests a novel method for learning representations of such networks.  The reviews are all in favor of acceptance, and they seem to favorably note the importance of this problem, as well as the novelty of the methods and the experiment outcomes.  I therefore am inclined toward accepting.